# Ecological Aspects of Distribution and Population Status Assessment of *Rhamnus erythroxyloides* subsp. *sintenisii* (Rich.f) Mabb., a Relict Species in the Kyzylkum Desert of Uzbekistan

**DOI:** 10.3390/plants13223154

**Published:** 2024-11-09

**Authors:** Khabibullo Shomurodov, Bekhzod Adilov, Alexander Rudov, Vasila Sharipova, Ozodbek Abduraimov, Rizamat Khayitov, Bekhruz Khabibullaev

**Affiliations:** 1Institute of Botany of the Academy of Sciences of the Republic of Uzbekistan, Durmon Yuli St. 32, Tashkent 100125, Uzbekistan; h.shomurodov@mail.ru (K.S.); bekhzod_a@mail.ru (B.A.); vasila_82@mail.ru (V.S.); ozodbek88@bk.ru (O.A.); bekh.xabibullaev@mail.ru (B.K.); 2Takhtajyan Institute of Botany NAS RA, Acharyan St. 1, Yerevan 0063, Armenia; 3Halophytes and C4 Plants Research Laboratory, Department of Plant Sciences, School of Biology, College of Sciences, University of Tehran, Tehran 14155-6619, Iran; 4Navoi State Pedagogical Institute, St. Ibn Sina St. 45, Navoi 210100, Uzbekistan; rizamathayitov368@ru.com

**Keywords:** rhamnaceae, desert shrubs, Kyzylkum, population ecology, remnant mountains, bioclimatic modeling, IUCN

## Abstract

*Rhamnus erythroxyloides* subsp. *sintenisii* (Rech.f.) Mabb. is a relict species occurring on the remnant mountains of Kyzylkum (in Uzbekistan). Up until recently, its population status and exact distribution remained unassessed. The ecological distribution and population status of four populations of *Rh. erythroxyloides* subsp. *sintenisii* were studied. The study of its morpho-anatomical structure revealed that the studied taxon’s vegetative and generative organs demonstrate clear adaptive features to arid climate. We observed an absence of the young fraction (seedlings and juvenile plants) in the populations, which is related to irregularity of seed renewal. Furthermore, population fragmentation due to a high degree of soil salinity and the presence of organic matter has been noticed at the present stage. According to the combination of threatening factors, the current state of the population of *Rh. erythroxyloides* subsp. *sintenisii* in Uzbekistan has been estimated as disappearing (EN)—B2ab (ii, iii, iv) + C1 + E.

## 1. Introduction

The remnant mountains of the Kyzylkum desert are isolated hills on a plain, elongated in the latitudinal direction, and are considered the western continuation of the Pamir-Alai Mountain system of Central Asia. The peculiar soil composition and climatic conditions determine the originality of their floristic composition, where a few mountainous Central Asian elements are still preserved (e.g., *Anemone petiolulosa* Juz., *Eranthis longistipitata* Regel, *Ranunculus sewerzowii* Regel, *Thalictrum isopyroides* C.A. Mey., *Tulipa undulatifolia* var. *micheliana* (Hoog) Wilford, *Lepyrodiclis holosteoides* (C.A. Mey.) Fisch. et C.A. Mey., *Prunus spinosissima* (Bunge) Franch.). The presence of these Central Asian mountain species, as well as a number of neoendemic and relict plant species, is the main difference between the Kyzylkum and the adjacent Karakum floras [1]. A relict species of the Kyzylkum desert is *Rhamnus erythroxyloides* subsp. *sintenisii* (Rich.f) Mabb., with a small population in the remnant mountains. Fossil remains of the genus *Rhamnus* L. are quite significant, with more than 50 species described in North America alone [2]. In Kazakhstan, there are samples from the Upper Cretaceous [3,4,5], and from the Paleogene and Neogene [6,7,8]. In addition, S.G. Zhilin emphasized the wide distribution of species of the genus *Rhamnus* on the territory of the Ustyurt plateau in the Upper Oligocene (Paleogene, Neogene) [9], which connects the origin of species of the genus *Rhamnus* with the Tertiary period. The genus *Rhamnus* L. has a wide distribution. Representatives of the genus grow on all continents except Australia. The genus has about 136 valid species [10,11]. In Central Asia there are seven species (*Rh. baldschuanica* Grubov, *Rh. cathartica* L., *Rh. dolichophylla* Gontsch., *Rh. grubovii* I.M.Turner, *Rh. songorica* Gontsch., and *Rh. erythroxyloides* subsp. *sintenisii* (Rich.f) Mabb.). Of these, five species (*Rh. baldshuanica*, *Rh. cathartica*, *Rh. songarica*, *Rh. Dolichophylla,* and *Rh. erythroxyloides* subsp. *sintenisii* (Rich.f) Mabb.) (Figure 1) are common in Uzbekistan. According to Plants of the World Online [11], the natural range of *Rh. erythroxyloides* subsp. *sintenisii* covers the territories of Morocco, Jordan, and, through Iran, it enters Central Asia (Turkmenistan and southwestern Kazakhstan, not reported for Uzbekistan) and Afghanistan. In Uzbekistan, it grows in the remnant mountains of the Kyzylkum: the Bukantau, Aktau, and Kuldzhuktau ridges. These areas are characterized by a Turanian climate type, with an activation of cyclones and intense precipitation in winter. The summer period is characterized by the formation of a Turanian continental tropical air mass, scorching heat, and prolonged dry weather. Modern research in the literature contains scanty data on the ecological and phytocenotic location of the subspecies. Based on the study of the flora and vegetation of the remnant mountains, P.K. Zakirov erroneously referred to *Rh. erythroxyloides* subsp. *sintenisii* (at that time the taxon status was equated to an independent species) as *Rhamnus coriacea* (Regel) Kom. (current *Rh. songorica* Gontsch.) [12]. L. Kapustina mentions the species as part of the *Rhamnus*—*Artemisia diffusa* and petrophytic shrub—*Artemisia diffusa*—*Oreosalsola arbusculiformis* associations [13], common on the northern slopes of the Tubabergan and Bozdontau mountains (Bukantau ridge). Analysis of data from the literature shows that *Rh. erythroxyloides* subsp. *sintenisii* (Rich.f) Mabb. previously has not been mentioned in any special research as a rare element of the Kyzylkum flora. Based on this, the purpose of this article is to assess the current state of the population of this taxon within Uzbekistan as a relict plant, the range of which is declining due to anthropogenic pressure.

## 2. Results

### 2.1. Phytocenotic Characteristics

To assess the current state of the populations of the study object within the three remnant ridges of the Kyzylkum desert (Bukantau, Aktau and Kuldzhuktau), four local populations were identified: BUT-1 (Bukantau Mountains 1), BUT-2 (Bukantau Mountains 2), KUT (Kuldzhuktau Mountains), and AKT (Aktau Mountains) (Figure 2). The four local populations are distributed within the following communities: *Krascheninnikovia ceratoides—Rhamnus erythroxyloides* subsp. *sintenisii* (BUT-1); *Krascheninnikovia ceratoides* + *Artemisia diffusa* + *Rhamnus erythroxyloides* subsp. *sintenisii* (BUT-2); *Rhamnus erythroxyloides* subsp. *sintenisii* + *Artemisia diffusa* + *Xylosalsola arbuscula* (KUT); and *Rhamnus erythroxyloides* subsp. *sintenisii* (AKT). In the above-mentioned plant communities, *Rhamnus erythroxyloides* subsp. *sintenisii* was rare—its projective cover varies from 3 to 8% (Appendix A). The flora of vascular plants of the remnant Kyzylkum Mountains differs significantly in species richness (including endemics) from other ecotopes (salt marsh, sand, and gypsum deserts) of this desert. According to Shomurodov et al., about 1050 species of flowering plants grow in the Kyzylkum [14], the basis of which are species common in the remnant mountains. Despite this, plant communities involving *Rh. erythroxyloides* subsp. *sintenisii* cannot be considered rich. They contain from 18 (BUT-2) to 29 (AKT) species of higher plants. While containing the largest number of species, the projective coverage of the communities of *Rh. erythroxyloides* subsp. *sintenisii* of the Aktau population was low—15%. The highest projective grass cover (22%) was noted in the Kuldzhuktau population (Appendix A). The remnant hills of Kulzhuktau, Aktau, and Bukantau are located in the latitudinal direction in Central Kyzylkum, which contributed to the formation of different types of vegetation, despite their location in the same climatic region in terms of aridity. The highest similarity index was observed between the Bukantau populations (73.7) (Table 1). The degree of difference in the floristic composition of plant communities is greater between the Kulzhuktau (KUT) and Bukantau populations. The community similar in floristic composition to the Kulzhuktau (KUT) population is Aktau (67.8). The formation of various types of plant communities in the study area is more related to edaphic than to climatic factors.

### 2.2. States of Local Populations

The results show that the age spectrum of the examined populations is not complete; in all coenopopulations (for details see Section 4.3), there is a lack of juvenile, and in BUT-1/BUT-2, also of immature individuals (Figure 3). In the latest local populations, the accumulation of the largest number of mature generative individuals was noted. They make up 34.4–38.5% of the total number of individuals. The ontogenetic spectrum of BUT-1/BUT-2 is centered with a predominance of mature generative individuals. In KUT, on the left side of the spectrum (im, v, g1) there is a smooth increase in individuals and their maximum value falls on old generative plants (34.1%). Unlike BUT-1/BUT-2, KUT and AKT are characterized by the presence of immature plants, although their proportion is insignificant (no more than 5.0%). The age spectrum of the KUT population is right-sided with a peak on old generative plants (g3), and the AKT population is left-sided, where the maximum values of individuals occur in young generative plants (g1). The average age spectrum of four coenopopulations shows that the number of individuals on the left side of the spectrum gradually increases and it reaches its maximum in the mature generative age of plants (g2) and then gradually decreases towards senescent plants (se) (Figure 4). A similar spectrum structure is typical for most shrubs in the arid zone.

### 2.3. Anatomical Structure of Vegetative and Generative Organs

The fruit of *Rhamnus erythroxyloides* subsp. *sintenisii* is a spherical succulent drupe with 3–4 seeds, is dark brown and shiny, and is 6–8 mm in diameter [15,16]. The pit is obovate, dark brown, triangular, convex on the dorsal side, and is 5 mm long. Seeds have a deep narrow groove on the dorsal side. The raphe is dorsal. The pericarp consists of three zones—exo-, meso-, and endocarp. The exocarp is leathery, the mesocarp is fleshy, and the endocarp is hard (Figure 5). The exocarp is thin and includes the epidermis and hypodermis. The epidermis consists of small, thick-walled cells. Under the epidermis is the hypodermis, which contains a brown mass (tannins). The mesocarp consists of thin-walled, relatively large-celled multirow parenchyma. The vascular bundles in the mesocarp are accompanied by 2–3 cells of hydrocyte tissue. Calcium oxalate druses are found in parenchymal cells. The endocarp forms the wall of the pit, which consists of an outer layer of stone cells and an underlying layer of sclerenchyma fibers. Between stone and sclerenchyma cells there are small crystal-bearing cells. Under the sclerenchyma fiber, there are thin-walled cells of the inner epidermis with a brown content (tannins).

The structural components of seeds are the embryo, endosperm, and seed coat. The seeds are exotestal. The exotesta is the main mechanical layer of the seed coat and is composed of prismatic thick-walled palisade cells with very narrow cavities. The peel is thin, tightly connected to the endocarp. The epidermis of the peel consists of stony cells. Beneath it are thin-walled parenchyma cells. At the bottom of the seminiferous groove, there is a conducting bundle of the seminiferous suture (raphe), enclosed in a cord of large thin-walled cells. The endosperm cells are thin-walled and contain dense storage substances—aleurone grains and oils. The embryo is erect, sometimes bent, often chlorophyll-bearing, differentiated into significantly predominant cotyledons, hypocotyl, and root. The leaves are dark green, alternate, oblong–oval, 1.6–2 cm long, 0.7–1 cm wide, with serrated edges. Both sides are covered with trichomes. The epidermis is single-row, large-celled, with a thickened outer wall. The mesophyll is isolateral-palisading (Figure 6). On the adaxial side, it is represented by 2–3 rows of palisade cells, on the abaxial side—1–2. Between them there are 4–5 rows of spongy cells, and the intercellular spaces are large. The central vein protrudes on the abaxial side. It contains one large collateral vascular bundle; the mechanical tissue is well developed in it. On both sides, groups of collenchyma cells are found under the epidermis. Around the vascular bundle, there are more rounded idioblasts. Calcium oxalate druses are found in the vein parenchyma and mesophyll. The lateral vascular bundles are weakly sclerified. The bark on young branches is reddish-brown and glossy. It is covered with multirowed periderm. In the bark, bast fibers are located in groups (Figure 6). The cambium multi-layered. The bulk of the wood is libriform. Libriform fibers are thick-walled, their cross-section is 5–6 sided, and the fiber lumens are round or oval. The radial rays are single-rowed, and their cells are narrow and thick-walled. The xylem ring is wavy in outline. At the boundary of the annual layers, there is a strip of cells compressed in the radial direction. In the wood, there are numerous vessels, and they are porous and spiral. The core is wide, and consists of large thin-walled cells, among which idioblasts and calcium oxalate druses are found.

### 2.4. Bioclimatic Modeling

High autocorrelation among several climatic parameters (bio_03, bio_05, bio_06, bio_09, bio_07, bio_10, bio_11, bio_13, bio_16, bio_17) led to their exclusion from further analysis, along with ten soil parameters, to avoid multicollinearity and enhance model accuracy. For both models, AUC values ranged from 0.9 to 1.0, indicating strong predictive power [17]. Specifically, the BE model achieved an AUC_training of 0.97 and an AUC_test of 0.94, while the BSE model demonstrated enhanced predictive accuracy with AUC_training = 0.99 and AUC_test = 0.97. Additionally, the 10% threshold for the BSE model was higher (0.564) compared to the BE model (0.384), confirming its robustness.

Key variables contributing to the BE model (≥10%) were precipitation of the warmest quarter (bio_18), temperature seasonality (bio_04), precipitation of the driest month (bio_14), and elevation (El_01). Jackknife analysis identified bio_18 as the most informative, showing the highest AUC alone and the lowest when excluded, indicating unique value. Other variables contributing less significantly, such as bio_02, bio_12, bio_01, bio_15, bio_19, and bio_08, are detailed in Appendix A.

In the BSE model, 45 of the 81 variables were retained, with significant contributions from gypsum-rich soils (S_07) and rocky substrates (S_41). Soil variable S_24 (secondary calcium carbonate accumulation) proved particularly informative, as did saline soils dominated by gypsum (S_17), both retaining unique information absent in other variables. For the BSE model, most variables exhibited high standard deviation (σ) values (Appendix A), as the variable values tended to deviate significantly from the mean (M), suggesting a wider distribution of variable values across study areas S1 and S2 [18].

The differentiation index (IMd) was especially notable for variables associated with soils that limit root development and show poor profile differentiation. High IMd values (≥1.0) for variables such as soils with secondary carbonates (S_24, S_59, S_64) and moderate salt accumulation indicate their essential role in defining favorable edaphic conditions for *Rh. erythroxyloides* subsp. *sintenisii*. Conversely, soils with Fe and Al behaviors or high organic matter in the upper horizon showed lower IMd values (≤1.0), signifying less influence on the ecological niche of the subspecies.

Both models (Figure 7) predict a limited distribution range for *Rh. erythroxyloides* subsp. *sintenisii* within 38–46° N and 48–72° E, covering northern Iran, western and northern Turkmenistan, southern and southwestern Kazakhstan, central and eastern Uzbekistan, and northwestern Kyrgyzstan. Key regions with optimal conditions include parts of the Alborz range, southwestern Kopet Dag, the Mangyshlak Peninsula, and the Karatau range.

The BSE model further extends the range across Central Asia, Iran, and Afghanistan (CAIA), including areas like western and eastern Karatau, northern and southern Aktau, and the Mangyshlak Peninsula, aligning with findings of N.I. Safronova (1996) [19]. These uplands, dating from the Middle Sarmatian, exhibit soils preserving Neogene limestone, chalk, and marl deposits [20,21]. Soils characterized by secondary carbonate (S_24) and sulfate (S_07) accumulation suggest habitat formation post-historic water recession. Additionally, *Rh. erythroxyloides* subsp. *sintenisii* is often associated with soils that limit root development, including dense clays (S_10), high-swelling clays, and rocky substrates (S_65). Soil salinity (S_17), originating from shallow groundwater, acts as a limiting factor in flat regions of the Turan Lowland, shaping the ecological constraints of the subspecies’ distribution.

### 2.5. Assessment of the State of the Population According to IUCN Criteria

As a relic of the Tertiary period, the current state of local populations of *Rh. erythroxyloides* subsp. *sintenisii* was assessed according to IUCN criteria. Currently, there are four local populations, clearly separated from each other. Although in recent years rare species have been regularly assessed in Uzbekistan based on the requirements of the IUCN Red List [22,23,24], this rare subspecies for Uzbekistan has not previously been included in the International Red List or national Red Book, so an assessment based on IUCN categories and criteria is being carried out for the first time. To date, a global assessment of the conservation status of the subspecies is impossible due to the lack of accurate data for this period on the state of populations of *Rh. erythroxyloides* subsp. *sintenisii* in Turkmenistan, Kazakhstan, and Iran, so we limited ourselves to conduct a regional assessment work. According to Geocat analysis, the distribution area of the subspecies in Uzbekistan (EOO) is 7224 km^2^ (Figure 8), and this indicator can serve as the basis for its compliance with the vulnerable (VU) category. However, the small area of local populations and the total number of individuals in them shows that the subspecies clearly corresponds to the endangered (EN) category. This is also confirmed by geospatial assessment: the total area (AOO) occupied by the population of the subspecies does not exceed 16 km^2^. The total number of individuals in the subpopulations is about 1000. The subpopulation in Aktau (about 500 individuals) has the largest proportion of individuals, while in Kuljuktau there are about 300 individuals, and in Bukantau (two subpopulations) about 200 individuals are registered, of which more than 90% are mature individuals. Analysis of general indicators in all subpopulations of *Rh. erythroxyloides* subsp. *sintenisii* shows that the AOO is less than 500 km^2^, there are no more than five localities, less than 2500 mature individuals in the population, and more than 20% risk of extinction in the next 100 years. All this indicates that this subspecies is on the verge of extinction ((EN) B 2ab (i, ii, iii, iv) + C1 + E) and the status is confirmed. Factors threatening the population (2.3 livestock and horse breeding, 5.2 collection of above-ground stems, 7.3 other ecosystem modifications [25]) are known, repeated, and not eliminated.

## 3. Discussion

*Rh. erythroxyloides* subsp. *sintenisii* is one of the rarest shrubs in Uzbekistan, common in the remnant mountains of the Kyzylkum desert (Bukantau, Aktau, and Kuldzhuktau ridges). Interestingly, it is not found in the Auminzatau mountains, which extend in parallel between the mentioned ridges, as well as in the Kazakhlitau and Kukchatau mountains, located in the north and northwest of the Kuldzhuktau ridge. This may be attributed to the fact that the last remnant mountains are not high and are not characterized by sharp rockiness, characteristic for this taxon. Populations of *Rh. erythroxyloides* subsp. *sintenisii* are characterized by self-maintenance of cenopopulations by seed propagation, a relatively short pre-generative (virginal) and post-generative (senile) period, and a long stay in a middle-aged generative state. These biological features suggest that the characteristic ontogenetic spectrum of cenopopulations of this taxon is centered. A centered spectrum, according to L.B. Zaugolnova [26], is formed in shrub plants with a long life expectancy of individuals in the middle-aged ontogenetic state, their low mortality, and difficult seed germination. According to the classification of A.A. Uranova and O.V. Smirnova, studied cenopopulations of *Rh. erythroxyloides* subsp. *sintenisii* are normal, but not full-membered [27]. The absence of juveniles in this coenopopulation is the result of irregular seed reproduction. This is probably related to environmental conditions of the habitat (the nature and humidity of the substrate and fluctuations in weather conditions). In years with low precipitation, flowers do not form in the vast majority of individuals. Another reason for the lack of seedlings is the washing out of seeds from the scree by precipitation. Thirdly, the most important aspect is the trampling of young, fragile individuals by farm animals. All described populations of the subspecies grow in an area of intensive grazing. The average age spectrum coincides with the theoretically established one and reflects the biological characteristics of this subspecies (Figure 3). However, the absence of both juvenile and immature individuals in the Bukantau populations (BUT-1, BUT-2) is an alarming indicator. In addition to the above factors, the cutting down of mature (g2 and g3) individuals by the local population as firewood also negatively affects the demographic state of the taxon (Figure 9). The anatomical structure of the leaves of *Rh. erythroxyloides* subsp. *sintenisii* bears characteristics of both typical xerophytes and mesophytes. Xeromorphic features include epidermis with thickened outer walls, the presence of trichomes, thick mesophyll of the isolateral palisade type, the presence of idioblasts, and sclerification of vascular bundles. Mesomorphic features include large-cell epidermis and large intercellular spaces. Adaptive characteristics of fruits and seeds are juicy drupes, dense pigmentation, sclerified pericarp, and the presence of calcium oxalate druses. The function of protecting the embryo is performed by a sclerified spermoderm and hydrocyte cells in the raphe. All of the above-mentioned traits, characteristic for many desert trees and shrubs, would contribute to the widespread distribution of this taxon throughout all the remnant mountains of Kyzylkum. However, it is found significantly scattered in a disjunctive range. The sporadicity of *Rh. erythroxyloides* subsp. *sintenisii* in the remnant mountains is not so much related to climatic and phytocenotic factors, but rather to edaphic factors. Within Uzbekistan, the main population of this taxon grows on calcareous substrates (therefore, some authors call it calcephile) [13]. Such substrates are more common in Bukantau and Aktau. According to data from the literature [28], *Rh. erythroxyloides* subsp. *sintenisii* is a plant of dry clay, gravelly, and rocky slopes of mountains and hills, growing on ravine cliffs, in rock cracks and on exposed limestone bedrock. The results of modeling the distribution of the subspecies indicate that this taxon is confined to Cretaceous bedrock deposits of uplands or residual rocks, the formation of which occurred after Paleogene time (Miocene) with the retreat of ancient seas [12,29,30,31,32]. According to O.E. Agakhanyants [33], *Rh. erythroxyloides* subsp. *sintenisii* is an inhabitant of extra-arid lowlands, covering the territories of Mangyshlak, Krasnovodsk Hills, Lesser Balkhan, and the remnant mountains of the Kyzylkum. The distribution of the taxon under study in Central Asia began in the Miocene, when, due to the extra-arid environment, salt-bearing sediments of gypsum and limestone were formed in the mountainous territories of this region [34]. Extra-arid uplands are places where soils were subject to less changes during the Quaternary period, which have consequently provided a habitat for *Rh. erythroxyloides* subsp. *sintenisii* [35].

## 4. Materials and Methods

### 4.1. Object and Area of Study

*Rh. erythroxyloides* subsp. *sintenisii* is a low, spreading, crooked, thorny shrub. It grows on southern rocky slopes, in rock crevices in the lower and middle mountain belt of predominantly limestone–dolomite mountains up to 1500 m [36]. The study area consists of the remnant mountains of the Kyzylkum Desert (Figure 2). The climate here is characterized by sharp fluctuations in annual temperatures, strong insolation, and a low precipitation. The bulk of precipitation falls in the winter–spring and partly in the autumn period (the maximum is observed from December to April). The annual precipitation is 70–125 mm. There is almost no precipitation in the summer [14,37]. Gray–brown soils are common on these remnant mountains [38,39].

### 4.2. Data Sampling

The work was carried out in 4 field seasons. In 2019, research was carried out in the Bukantau ridges, in 2020 and 2022 the Kuljuktau population was studied, and in 2021—the Aktau population (Figure 2). To determine the location of local populations of *Rh. erythroxyloides* subsp. *sintenisii*, the author’s field data obtained in 2011–2021, herbarium materials stored in TASH, and literary sources were used [12,13]. To establish structural and adaptive characteristics, samples of *Rh. erythroxyloides* subsp. *sintenisii* were collected from the Bukantau region. For anatomical research, samples were taken from 5 plants of the middle tier, which were in flowering–fruiting phase. The initial data for the distribution modeling of *Rh. erythroxyloides* subsp. *sintenisii* were presence-only data. The analysis included only data confirmed by verified herbarium specimens.

The analysis of the taxon’s distribution range was carried out on the basis of herbarium material from the National Herbarium of Uzbekistan (TASH) and stock materials of the Global Biodiversity Database [40]. In addition, the analysis includes our own field data obtained during the period 2011–2021. The search for coordinates of geographical points of herbarium specimens was carried out using SAS Planet19 based on topographic maps. As a result, 21 localities of *Rh. erythroxyloides* subsp. *sintenisii* were identified. in Central Asia, Iran, and Afghanistan. All localities were processed by spatially filtering the subspecies location data to maximize the number of spatially independent locations using SDMtoolbox [41,42].

### 4.3. Data Analysis

Population studies were based on the concept of plant coenopopulations, which is a complex biosystem consisting of individuals of different age states [43,44]. The set of age groups and their numbers determine the age structure of coenopopulations (coenopopulation is a set of individuals of a taxon in a certain phytocenosis) [45]. In the case when an accurate determination of the age of individuals was impossible, the population was characterized by the ratio of individuals included in different age groups, and in this case, we speak not about the age state, but about the age spectrum of the population. In this work, to assess the state of the populations of the taxon under study, we used the age spectrum of the coenopopulations of the taxon.

The ontogenetic state of individuals was determined using a method developed for woody plants [27,46] ratio of individuals of different ontogenetic states in the coenopopulation. An individual was taken as a counting unit. The study of population structure was based on the idea of a characteristic ontogenetic spectrum [25]. Coenopopulations were described according to the classifications of A.A. Uranova and O.V. Smirnova [47]. Counts were carried out on temporary sample plots of 10 × 10 m. The number of sample plots in the community was selected based on the degree of its horizontal heterogeneity and the need to take into account individuals of all ontogenetic groups represented in the community. To establish the similarity of the floristic composition of the communities in which local populations of the taxon under study were studied, the Sørenson index was used [48]:CC = 2c/(a + b + 2c)
where:c—number of similar species between communities;a and b—the number of dissimilar species between communities.

An anatomical study of the structure of the leaf blade, branches, and fruits of *Rh. erythroxyloides* subsp. *sintenisii* was performed in order to identify key adaptive traits to drought conditions, such as a thickened outer epidermal wall, the presence of trichomes and leaf idioblasts, and a dense, pigmented, sclerified pericarp of the fruit. At the same time, the morphology of fruits and seeds is described according to the generally accepted method developed by N.N. Kaden and E.S. Smirnova [49]. Fruits and vegetative organs were fixed in 70% ethanol. Anatomical studies were carried out according to Barykina and Chubatova [50]. The fruits were kept for a month in a Strassburger–Flemming solution: alcohol, glycerin, water (1:1:1). Transverse sections of leaf blades, branches, and fruits were made freehand with a razor. The study of preparations and micrographs were carried out on a KERN OBN 1327241 plan achromatic microscope. Bioclimatic modeling was performed using the maximum entropy method implemented in the MaxEnt program [51]. The modeling of the distribution of the subspecies relied on two models:(1)BE model. When constructing the models, 19 bioclimatic parameters (Bioclim) (Appendix A) were used, reflecting data on temperature and precipitation of the territory [52], which allow for the interpolation of observed data from 1979 to 2013 [53]. The model also includes elevation data parameters. Bioclimatic information was obtained from the Worldclim database [54] at a resolution of 30 arc seconds. Based on the Global Digital Relief Model (DEM) [55], elevation data with a resolution of 90 m were constructed.(2)BSE model. The model consists of 19 bioclimatic, 82 soil parameters, and altitude data (Appendix A). Soil data were based on the World Soil Resources Reference Base (WRB) and included 4 soil horizon characteristics, 5 physical and 4 soil chemical properties, and 69 soil classifications of the area of CAIA at 250 m resolution. Soil groupings derived from their reference soil groups [56].

To assess the contribution of all variables in the models, autocorrelation of the variables in each model was performed, as a result of which variables whose correlation did not exceed 0.8 were included in further analysis. In addition, the main statistical parameters of the variables (values of minimum (min), maximum (max), mean (M), and standard deviation (σ)) were obtained based on the SDM Toolbox [42,57]. When building the model in the MaxEnt environment, the Bootstrap replication method with 5000 maximum interactions was used to train the data. To test the accuracy of the model, 25% of the points were used as a testing sample. A threshold of 10 percentiles was also set, which provided for the exclusion from the process of 10% of points located in extreme climatic conditions for the species. They were considered as growing in conditions atypical for the species and were not taken into account when constructing the rasterization of the model. We chose a logistic format with gradations from 0 to 1 as the output format to estimate the probability of species findings.

Distribution modeling results of *Rh. erythroxyloides* subsp. *sintenisii*, were analyzed in two aspects—on the territory of CAIA (S1) and on the territory of the econiche (S2) of the subspecies based on the degree of accumulation M value soil classification variables, whose total contribution to building the model was greater than others. To obtain information about the change in the interval M value of variables in S1 after modeling, we used the differentiation index IMd (S2M/S1M), the results of which helped to extract data from the most important soil factors in the territory of the taxon’s econiche (S2). A scatterplot using Origin Pro 8 was used to determine the IMd weight. When compiling species distribution maps, GIS packages ArcMap 10.8, QGIS 3.10.2, and DIVA-GIS 7.4.0.1 were used. The degree of endangerment of populations and the rarity status of *Rh. erythroxyloides* subsp. *sintenisii* were identified using IUCN criteria [25], a methodology based on population numbers (species range, number of adults, local population size) and assessment of their trends over time. The species’ distribution area (EOO—extent of occurrence) and the area occupied by its populations (AOO—area of occupancy) were calculated using the Geospatial Conservation Assessment Tool (GeoCAT) program [58].

## 5. Conclusions

Isolation in the remnant mountains and limitation of the spread of *Rh. erythroxyloides* subsp. *sintenisii* in the flats of CAIA is associated with the predominance of saline habitats and the active accumulation of organic matter at the present stage of soil formation. In this context, the fragmented Cretaceous bedrock habitats of the remnant uplands of the Kyzylkum Desert provide a unique location for the conservation of its populations. Despite the relatively wide distribution of *Rhamnus* throughout the world, the population of the *Rh. erythroxyloides* subsp. *sintenisii* in Uzbekistan is highly fragmented and distributed in a narrow range. Local populations of *Rh.erythroxyloides* subsp. *sintenisii* in Uzbekistan are distributed only in the remnant mountains of the Kyzylkum desert, located in its northern and central parts. Based on this, it can be assumed that the risk of extinction of the taxon under study in the next 100 years is at least 20% (E). In general, taking into account all the above factors, the current state of the population of *Rh. erythroxyloides* subsp. *sintenisii* in Uzbekistan can be assessed as endangered (EN)—B2ab (ii, iii, iv) + C1 + E and the subspecies is recommended for inclusion in the IUCN Red List and the Red Book of the Republic of Uzbekistan.

## Figures and Tables

**Figure 1 plants-13-03154-f001:**
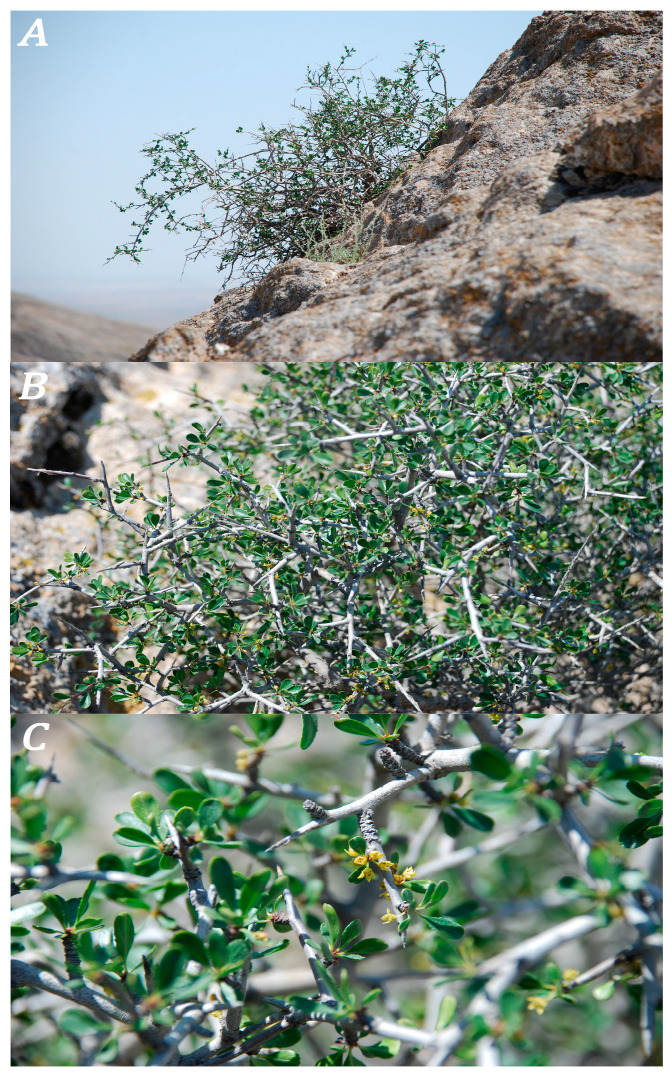
*Rhamnus erythroxyloides* subsp. *sintenisii*: (**A**) general appearance of the plant and its habitat; (**B**) dense branched stem; (**C**) generative branches.

**Figure 2 plants-13-03154-f002:**
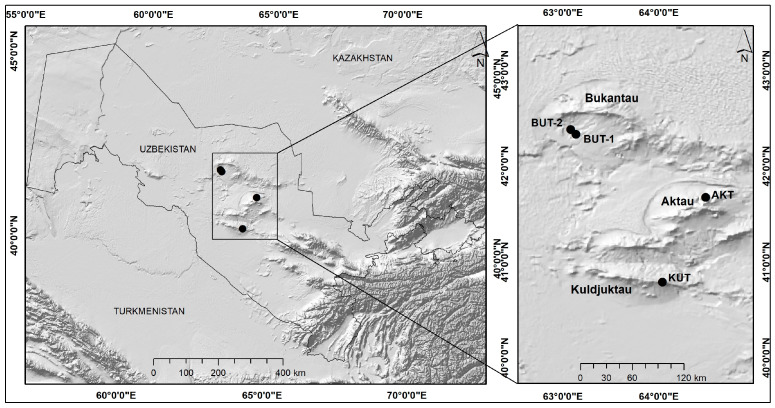
Study area. BUT-1, BUT-2 (Bukantau Mountains), KUT (Kuldzhuktau Mountains), and AKT (Aktau Mountains).

**Figure 3 plants-13-03154-f003:**
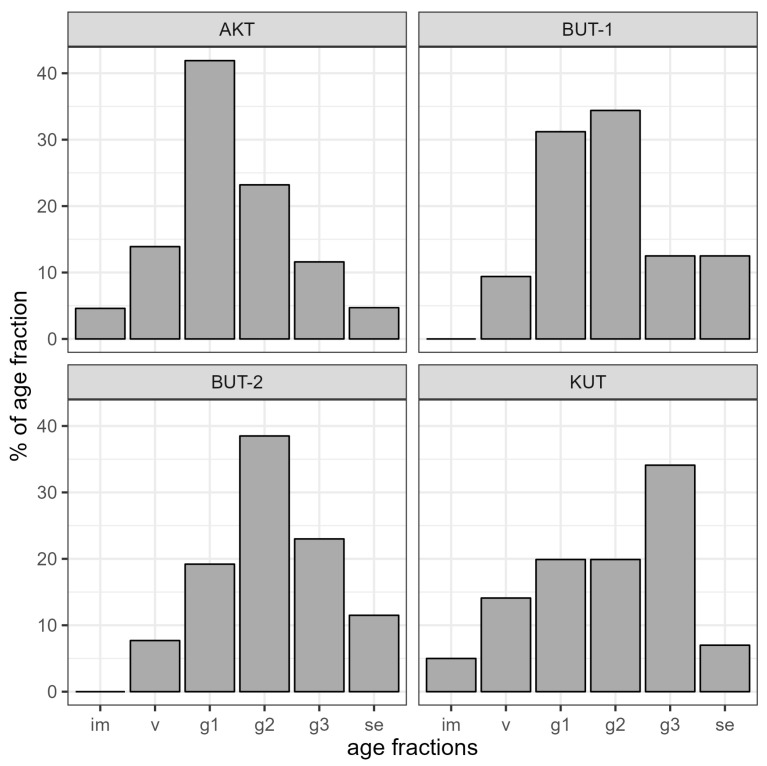
Age structure of local populations of *Rhamnus erythroxyloides* subsp. *sintenisii* (im = immature plants, v = virginal plants, g1 = young generative plants, g2 = mature generative plants, g3 = old generative plants, se = senescent plants).

**Figure 4 plants-13-03154-f004:**
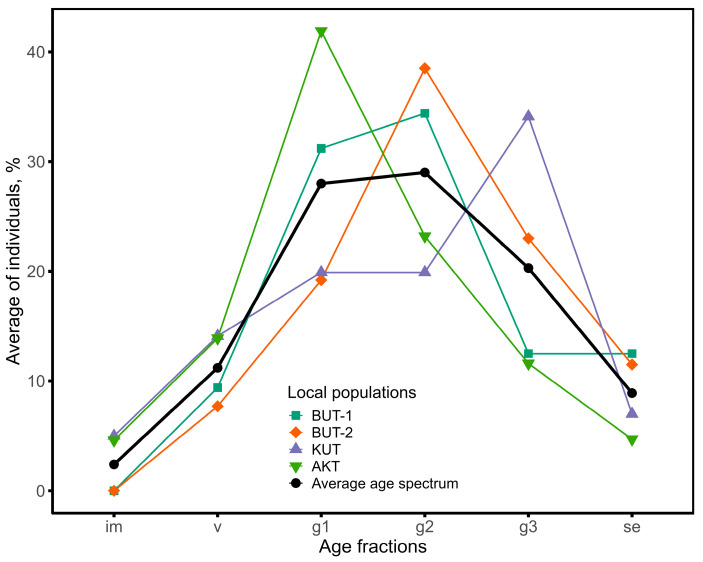
The average age spectrum of populations of *Rhamnus erythroxyloides* subsp. *sintenisii*.

**Figure 5 plants-13-03154-f005:**
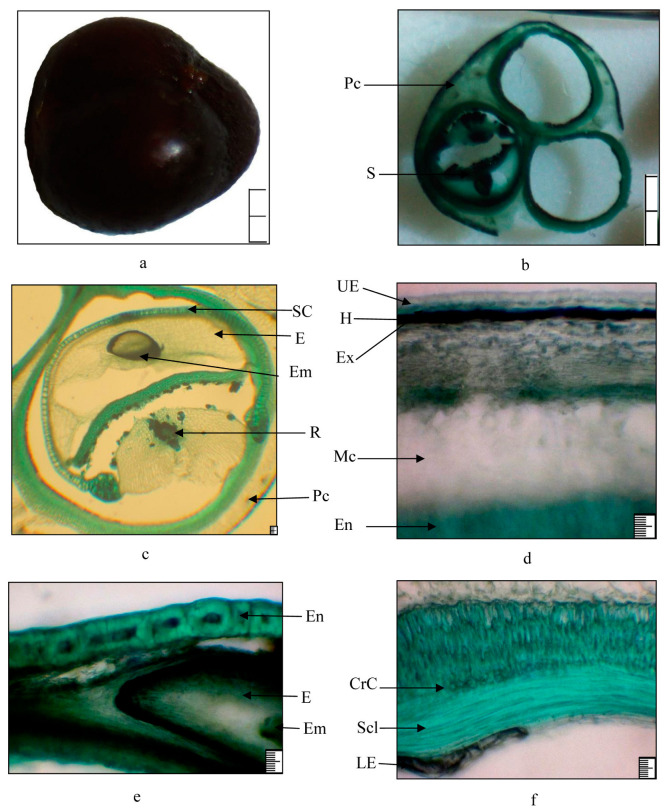
Structure of the fruit of *Rhamnus erythroxyloides* subsp. *sintenisii* i: (**a**) general view; (**b**) cross section of the fruit; (**c**) seed; (**d**,**e**) fragments of the pericarp; (**f**) fragment of seeds. Legend: CrC—crystalline cells, E—endosperm, Em—embryo, En—endocarp, Ex—exocarp, H—hypodermis, LE—lower epidermis, Mc—mesocarp, Pc—pericarp, R—raphe, S—seed, SC—seed coat, Scl—sclerenchyma, UE—upper epidermis; (**a**,**b**) 20 mm, (**c**–**f**) 100 µm.

**Figure 6 plants-13-03154-f006:**
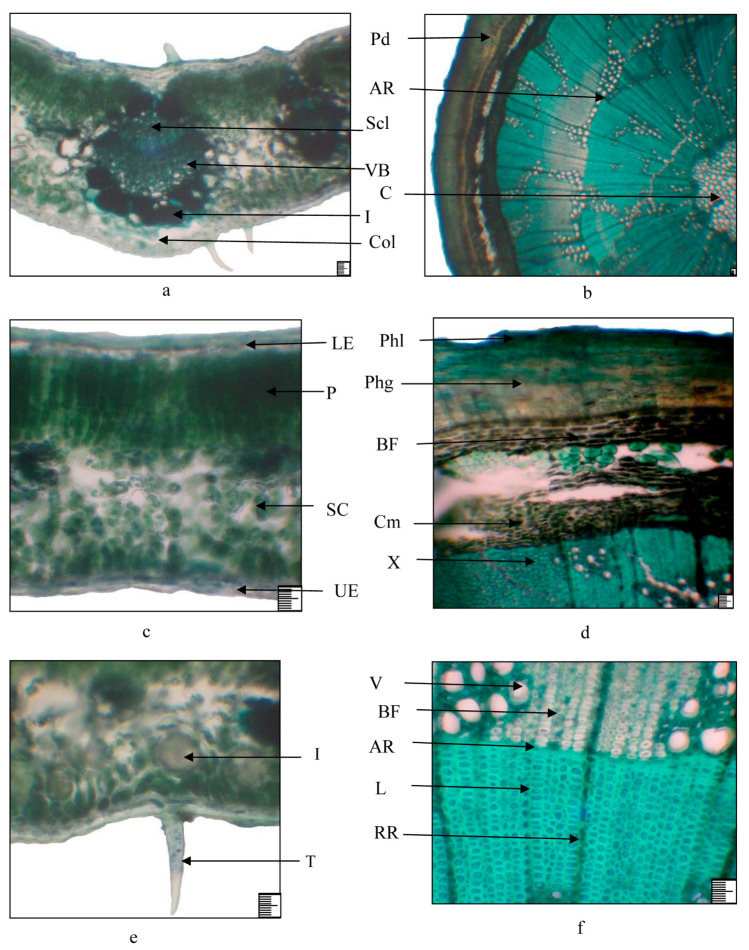
Leaf and branch structure of *Rhamnus erythroxyloides* subsp. *sintenisii* i: (**a**,**c**,**e**) leaf; (**b**,**d**,**f**) branch. (**a**) Main vein; (**c**,**e**) mesophyll; (**b**) general view of a cross section of a branch; (**d**) bark fragment; (**f**) fragment of xylem. Legend: AR—annual growth rings, BF—bast fibers, C—core, Cm—cambium, Col—collenchyma, I—idioblast, L—libriform, LE—lower epidermis, P—palisade, Pd—periderm, Phg—phellogen, Phl—phellem, RR—radial ray, SC—spongy cells, Scl—sclerenchyma, T—trichome, V—vessel, VB—vascular bundle, UE—upper epidermis, X—xylem; (**a**–**f**) 100 µm.

**Figure 7 plants-13-03154-f007:**
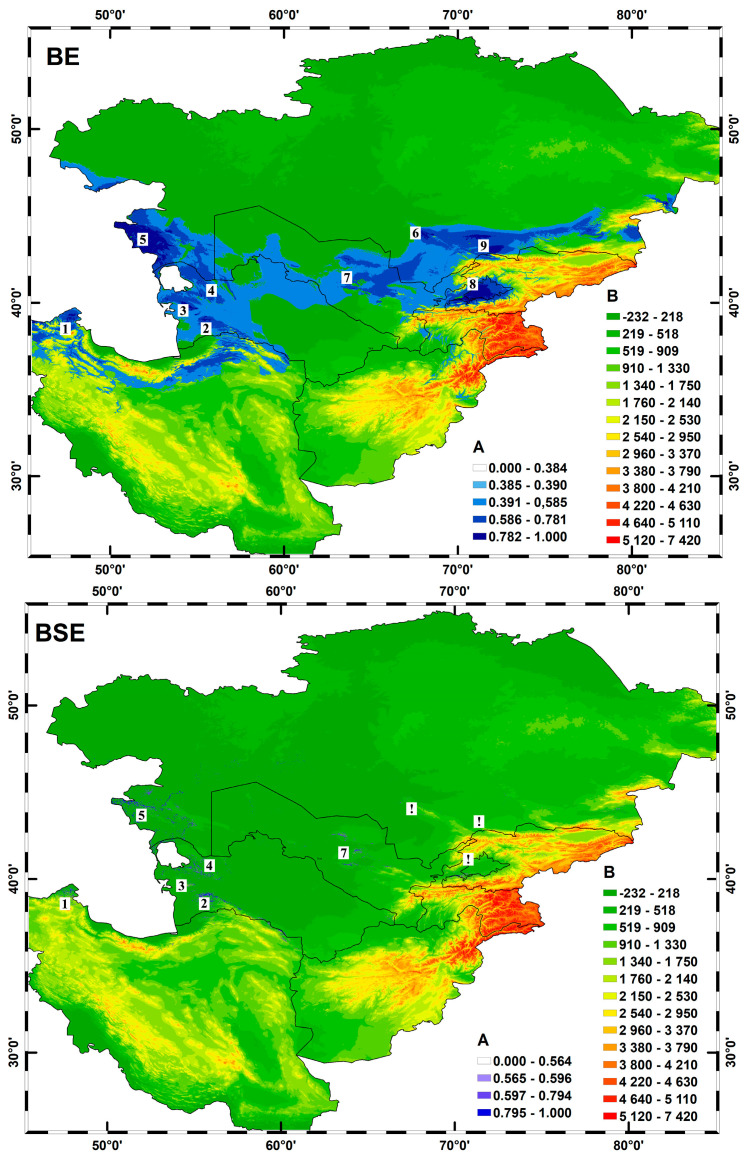
Potential ranges of *Rh. erythroxyloides* subsp. *sintenisii* based on BE and BSE models: A—assessment of the degree of distribution of the species; B—height above sea level (m). According to potential hotspots, *Rh. erythroxyloides* subsp. *sintenisii* is distributed across the northernmost part of the Elburz Ridge (1) (Iran), partially extending to the southwestern slopes of the Turkmen–Khorasan Mountains—Kopet Dag (2), the Small and Large Balkhan Mountains (3), and the Krasnovodsk Hills (4) (Turkmenistan). Its range also includes the expansive Mangyshlak Peninsula (5) and the northwestern slopes of the Karatau Ridge (6) (Kazakhstan), certain remnant mountains of the Kyzylkum Desert (7), the extensive Fergana Valley (8) (Uzbekistan), and the northern and western slopes of the Kyrgyz Ridge (9) (Kyrgyzstan). However, based on the BSE distribution model, the occurrence of this species in the Karatau Ridge (6), Fergana Valley (8), and Kyrgyz Ridge (9) is unlikely.

**Figure 8 plants-13-03154-f008:**
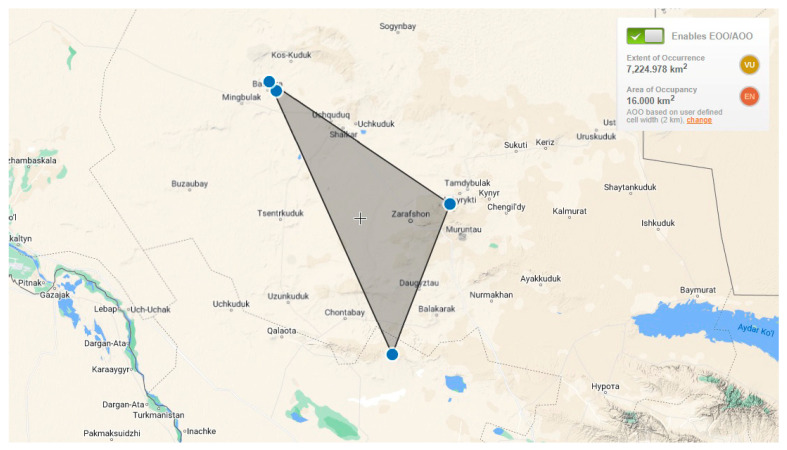
GeoCAT map of *Rhamnus erythroxyloides* subsp. *sintenisii* in Uzbekistan.

**Figure 9 plants-13-03154-f009:**
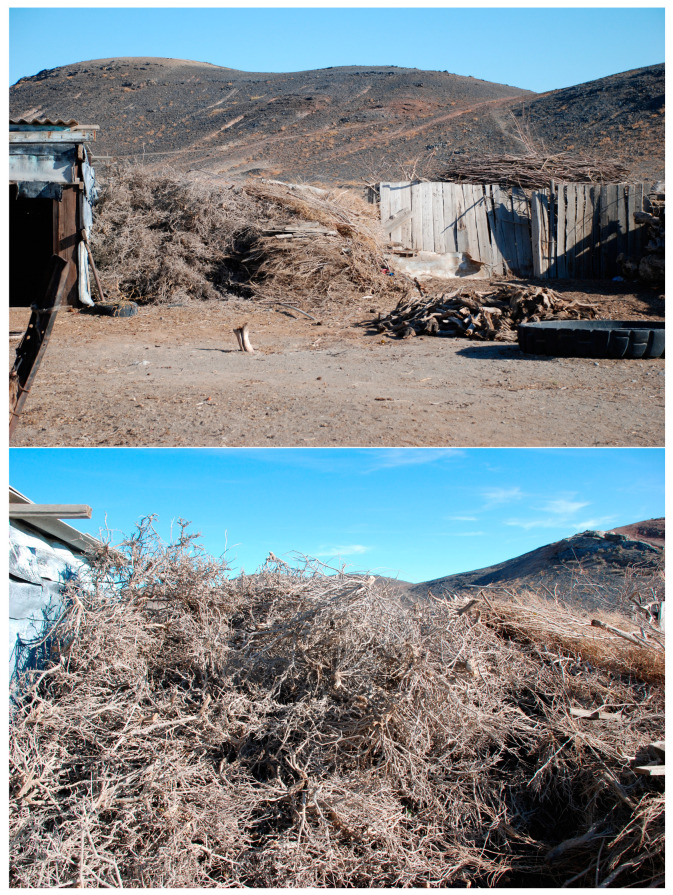
Harvested firewood from large specimens of *Rh. erythroxyloides* subsp. *sintenisii* by residents of the village of Zharkala, located next to the Bukantau populations.

**Table 1 plants-13-03154-t001:** Analysis of the similarity of plant communities according to the Sørenson index.

AKT	KUT	BUT-2	BUT-1	Population
			1	BUT-1
		1	73.7	BUT-2
	1	41	34.8	KUT
1	67.8	50	44	AKT

## Data Availability

The datasets generated during and/or analyzed during the current study are available from the corresponding author upon reasonable request.

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
