# Peer review of "Ecological Aspects of Distribution and Population Status Assessment of Rhamnus erythroxyloides subsp. sintenisii (Rich.f) Mabb., a Relict Species in the Kyzylkum Desert of Uzbekistan"

_plants, 2024, doi:10.3390/plants13223154_

Round 1
Reviewer 1 Report
Comments and Suggestions for Authors
Overall, a solid contribution but could be tightened up in the text (make shorter and more precise) as well as make corrections in English usage as noted seperately. Also recommend defining some terms which are more prevelant in Russian botanical usage than in western usage (ie., cenopopulations)
For example, I recommend in the section on Range Modelling, just to discuss the core variables, put other items into a table or supplemental table.
Some other technical items to revise
Page 6-on Figure 4, label different lines and the median line (currently no labels on lines)
Page 9, line 225. Define IMd
Page 9, line 241 Define CAIA here (it is defined later in the methods but this is the first occurrence in the paper)
Page 9, line 246. Citation to year should be changed to numerical citation Safranova 1996
Page 11, lines 309, 310: The terminology cenopopulations and pregenerative are not commonly used; replace generative with reproductive; define cenopopulation
In Supplemental table 1, define what + and - mean in title or footnotes.
Comments on the Quality of English LanguageThe following are recommendations to improve the quality of the English used: I cite the page and line where the recommendation applies.
Page 1, line 17: In the Abstract, Delete "Within a three-year long study", add at the end of the sentence "over a three-year period"
Page 1, line 26. In Keywords: the meaning of remnant mountains is unclear; I believe it refers to an isolated chain, perhaps consider using this terminology instead
Page 2, line 49: Italicize the specific epithet "baldschuanica"
Page 4, line 113; change juvenile to juveniles
Page 5, line 126; Unclear meaning of "cyanide plants" Rewrite or delete
Page 6, line 143: typo-change forma to forms
Page 9, line 209. typo-Change Analyzes to Analysis
Page 9, line 253: unclear meaning of "breeds" rewrite or delete
Page 11, line 314; unclear meaning of "their least elimination"; rewrite
Page 12, line 320; rewrite "generative organs" to "flowers"
Page 12, line 337: typo raff change to raphe
Page 14, line 428, put space between subspecies and relied
In Supplemental Table 2. Misspelling noted in "Soil caraterized by the dynamics of Fe and Al" should read "Soil characterized by the dynamics of Fe and Al"
Author Response
Dear Reviewer, thank You so much for Your precious work and patience and Your important
improvements!
Here is a short list of responses to Your comments.
Comment 1: Overall, a solid contribution but could be tightened up in the text (make shorter and more precise) as well as make corrections in English usage as noted seperately. Also recommend defining some terms which are more prevelant in Russian botanical usage than in western usage (ie., cenopopulations)
Response: We worked on the English, both considering Your notes and changing minor issues in the text. Considering the term "coenopopulation" there is a short explanation in te methods section and we added a reference to that section where the term appears the first time.
Comment 2: For example, I recommend in the section on Range Modelling, just to discuss the core variables, put other items into a table or supplemental table.
Response: WE shortened and revised this section and hope in its current worm it is easier to comprehend.
Comment 3: Page 6-on Figure 4, label different lines and the median line (currently no labels on lines)
Response: edited
Comment 4: Page 9, line 225. Define IMd
Response: edited name - definition appears also in the methods
Comment 5: Page 9, line 241 Define CAIA here (it is defined later in the methods but this is the first occurrence in the paper)
Response: edited
Comment 6: Page 9, line 246. Citation to year should be changed to numerical citation Safranova 1996
Response: edited
Comment 7: Page 11, lines 309, 310: The terminology cenopopulations and pregenerative are not commonly used; replace generative with reproductive; define cenopopulation
Response: we accordingly added a link to the methods in the place the word coenopopulation appears for the first time and added some more clear explaining terms
Comment 8: In Supplemental table 1, define what + and - mean in title or footnotes.
Response: editedComment 9: The following are recommendations to improve the quality of the English used: I cite the page and line where the recommendation applies.
Response: edited ecept for - see comment 10
Comment 10: Page 1, line 26. In Keywords: the meaning of remnant mountains is unclear; I believe it refers to an isolated chain, perhaps consider using this terminology instead
Response: This term appears also in English literature and is also already specifically used for the mountain chains of the Kyzylkum desert. It refers rather to the geological origin of the mountain chains than to their geographical isolation -thus, we would suggest to keep this term if possible! It is already established not only as a general term but also more specifically to the mountain chains of the study area. If necessary, we may change it , but we bielieve this term explains better the specifiic conditions of these mountain chains.
Reviewer 2 Report
Comments and Suggestions for Authors
The manuscript "Ecological aspects of distribution and population status assessment of Rhamnus erythroxyloides subsp. sintenisii (Rich.f) Mabb., a relict species in the Kyzylkum desert of Uzbekistan" is certainly interesting, dealing with a species that appears to be in serious danger of extinction.
Despite this, there are serious problems with the writing that should be corrected before its eventual publication.
In terms of data collection, only the 4 surveys shown in Supplementary Table 1 were conducted in the study area? And, if percentages are reported in the table (in English, decimals are separated from units by the use of a period), what do the plus symbols mean?
Is there so much difference between the BUT1 and BUT2 populations that they should be considered separate? They seem very close to me and very similar in floristic terms, what features differentiate them apart from what appears to be a different occurrence of halophilic species?
You need to check the species names in the text (some are misspelled and some times italics are missing) and in Supplementary Table 1 (some are misspelled).
line 128: the caption in Figure 3 lacks the legend for the abbreviations (im, v, g1, g2, g3, and se), which are also not explained in the data and methods section
line 131: in the caption of Figure 4, the line-dot legend is missing
line 143: is the Latin word "forma" intentional?
line 269: in the caption of figure 7, the legend of the numbers in the squares is missing
line 410: "[48]." should be "[48]:"
line 479: "20% (E)" should be "20% (E)."
Comments on the Quality of English LanguageI am not an expert in English, but the text flows quite well, albeit with a few more convoluted points.
Author Response
Dear Reviewer, thank You so much for Your precious work and patience and Your important
improvements!
Here is a short list of responses to Your comments.
Comment 1: In terms of data collection, only the 4 surveys shown in Supplementary Table 1 were conducted in the study area? And, if percentages are reported in the table (in English, decimals are separated from units by the use of a period), what do the plus symbols mean?
Response: We accordingly added a short explanation to address this issue
Comment 2: Is there so much difference between the BUT1 and BUT2 populations that they should be considered separate? They seem very close to me and very similar in floristic terms, what features differentiate them apart from what appears to be a different occurrence of halophilic species?
Response: Thank You for addressing the issue. Indeed both populations are close in the terms of species composition and geographically, they however substantially differ in the dominant species, degree of degradation, as You mentioned the presence of halophytes as well as exposition, thus we considered them separately.
Comment 3: You need to check the species names in the text (some are misspelled and some times italics are missing) and in Supplementary Table 1 (some are misspelled).
Response: done
Comment 4: line 128: the caption in Figure 3 lacks the legend for the abbreviations (im, v, g1, g2, g3, and se), which are also not explained in the data and methods section
Response: done
Comment 5: line 131: in the caption of Figure 4, the line-dot legend is missing
Response: done
Comment 6: line 143: is the Latin word "forma" intentional?
line 269: in the caption of figure 7, the legend of the numbers in the squares is missing
line 410: "[48]." should be "[48]:"
line 479: "20% (E)" should be "20% (E)."
Response: done